# Energy–speed relationship of quantum particles challenges Bohmian mechanics

Violetta Sharoglazova[1], Marius Puplauskis[1], Charlie Mattschas[1], Chris Toebes[1] & Jan Klaers[1✉]

Classical mechanics characterizes the kinetic energy of a particle, the energy it holds due to its motion, as consistently positive. By contrast, quantum mechanics describes the motion of particles using wave functions, in which regions of negative local kinetic energy can emerge[1]. This phenomenon occurs when the amplitude of the wave function experiences notable decay, typically associated with quantum tunnelling. Here, we investigate the quantum mechanical motion of particles in a system of two coupled waveguides, in which the population transfer between the waveguides acts as a clock, allowing particle speeds along the waveguide axis to be determined. By applying this scheme to exponentially decaying quantum states at a reflective potential step, we determine an energy–speed relationship for particles with negative local kinetic energy. We find that the smaller the energy of the particles—in other words, the more negative the local kinetic energy—the higher the measured speed inside the potential step. Our findings contribute to the ongoing tunnelling time debate[2–6] and can be viewed as a test of Bohmian trajectories in quantum mechanics[7–9]. Regarding the latter, we find that the measured energy–speed relationship does not align with the particle dynamics postulated by the guiding equation in Bohmian mechanics.

Quantum particles can be found in regions in which classical particles are not allowed to exist. An example of this is evanescent (exponentially decaying) quantum states at potential barriers and steps. When a stream of particles encounters a potential step that is reflective from a classical viewpoint, there is a non-vanishing probability of finding particles within the potential step. This intriguing behaviour arises from the ability of quantum mechanical states and their associated wave functions $\psi(\mathbf{x})$ to generate domains of negative kinetic energy[1], where $\psi(\mathbf{x})^{-1}\hat{\mathbf{p}}^2\psi(\mathbf{x})/2m < 0$. Here, $\hat{\mathbf{p}}$ and $m$ denote the momentum operator and mass, respectively, and $\psi(\mathbf{x})$ is considered an energy eigenstate. This ability enables quantum particles to migrate into regions considered energetically forbidden in classical terms while maintaining energy conservation. Although this is well known, the kinematic interpretation of these states raises questions that have not yet been conclusively answered. This includes questions about whether a speed can be assigned to these states of motion, and if so, how it depends on energy.

The kinematic interpretation of classically forbidden states of motion is intrinsically linked to the longstanding problem of tunnelling times. This problem refers to the question of how long a particle needs to tunnel through a (finite) potential barrier[2–6]. Despite the controversy surrounding this topic, important insights, methodologies[10–18] and experiments[19–27] have emerged from this debate. Investigating energy–speed relationships in evanescent phenomena can contribute to the discussion by offering a complementary perspective on this problem. Speed measurements in quantum mechanics can also serve as tests for the Bohmian interpretation of quantum mechanics, in which particle velocities play a very prominent part[7,28–30]. Apart from the Schrödinger equation, a guiding equation is assumed that assigns a velocity field $\mathbf{v}_S(\mathbf{x}, t)$ to a wave function $\psi(\mathbf{x}, t) = \sqrt{n(\mathbf{x}, t)}\exp(iS(\mathbf{x}, t))$ using

$$\mathbf{v}_S(\mathbf{x}, t) = \frac{\hbar}{m}\nabla S(\mathbf{x}, t). \tag{1}$$

Here, the real-valued functions $n(\mathbf{x}, t)$ and $S(\mathbf{x}, t)$ describe the particle density and phase of the wave function, respectively. The velocity field $\mathbf{v}_S(\mathbf{x}, t)$ gives rise to trajectories along which the particles are expected to move. If the initial particle positions of these trajectories are chosen according to the Born rule (quantum equilibrium hypothesis), the resulting probability density aligns with the predictions of standard quantum mechanics. Whether this implies that both theories generally make identical predictions remains a matter of debate. Discrepancies are sought, for example, in temporal quantities such as tunnelling or arrival times[31–34]. Although measurement results in standard quantum mechanics are considered inherently random, the existence of Bohmian mechanics—a fully deterministic dynamical system—demonstrates that quantum phenomena do not necessarily rely on randomness at their core. These sharply divergent views on the fundamental processes of nature motivate the search for experimental tests to distinguish between the two theories.

In our experiments, we investigate states of motion that arise when a particle is reflected from a potential step. Standard quantum mechanics does not incorporate particle velocities into its formalism in the same explicit way as Bohmian mechanics. Moreover, conventional measures of motion—such as group or phase velocity—do not yield physically meaningful results in this scenario. To measure particle speeds associated with these states, we compare translational motion

[1]Adaptive Quantum Optics, MESA+ Institute of Nanotechnology, University of Twente, Enschede, The Netherlands. ✉e-mail: j.klaers@utwente.nl

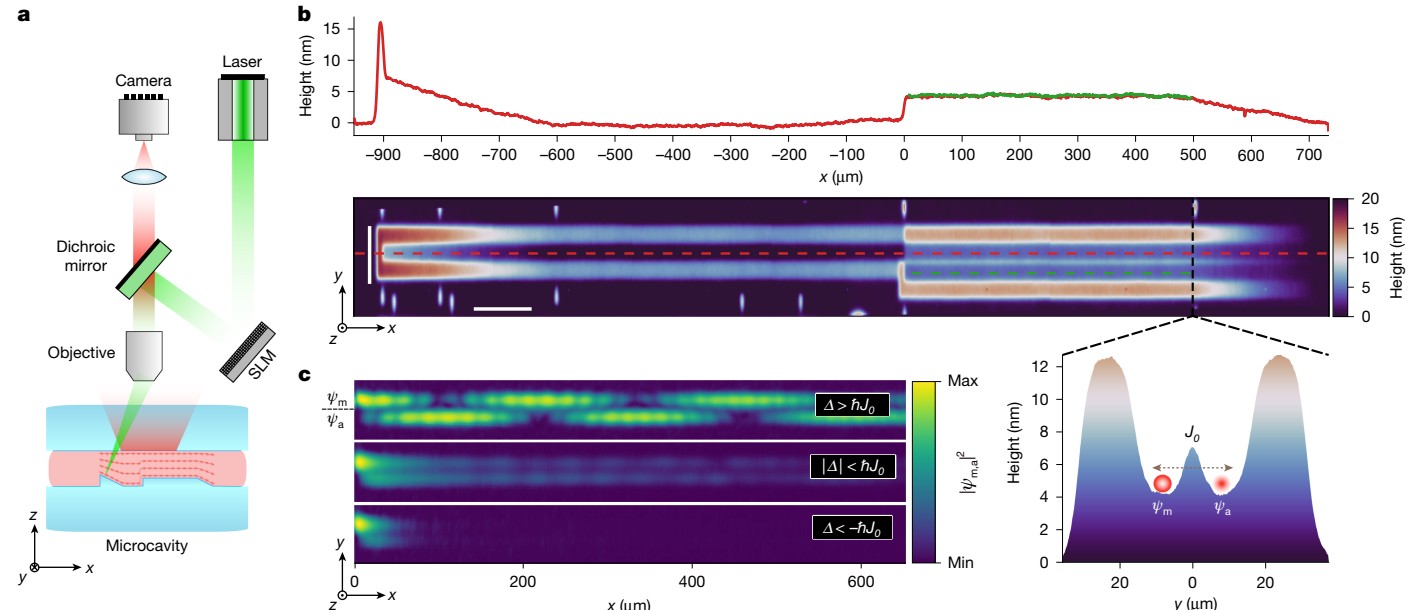

**Fig. 1 | Experimental setup for measuring the speed of particles. a**, Schematic of the experimental setup. The microcavity consists of two planar mirrors, one of which is nanostructured, with an optically active dye in between. The optical medium can be non-resonantly pumped to induce a lasing process, generating microcavity photons at the point of pumping. A small amount of the light circulating in the microcavity is transmitted through the mirrors and can be imaged onto a camera. **b**, Height map and cross-sections (see colour coding) of the nanostructured mirror. The height profile of the nanostructured mirror effectively induces a potential energy landscape that confines and guides the photons in the transverse plane of the resonator. Specifically, the light is guided in a waveguiding potential from $x = -900$ µm to the end of the structure at $x = 500$ µm. In the first section, from $x = -900$ µm to $x = -600$ µm, a linear ramp potential is superimposed on the waveguide potential. By changing the position of the non-resonant optical pumping along the ramp with a spatial light modulator (SLM), we can set the initial potential energy of photons. At $x = 0$, a step potential of height $V_0 = (0.538 \pm 0.003)$ meV is superimposed on the waveguide that runs from $x = 0$ µm to $x = 500$ µm. Apart from the main waveguide described above (red cross-section), an auxiliary waveguide (green cross-section) is introduced at the start of the step potential at $x = 0$. This effectively creates a double-well potential in the direction orthogonal to the waveguide axis (grey cross-section), with a coupling between the respective ground states of the wells of $J_0 = 2\pi(6.34 \pm 0.01)$ GHz. **c**, Camera images showing the photon populations in the coupled waveguides for three distinct energy regimes associated with three different dynamics: population oscillations in the classically allowed regime ($\Delta > \hbar J_0$), long-range non-oscillatory propagation ($|\Delta| \leq \hbar J_0$) and evanescent decay ($\Delta < -\hbar J_0$). Scale bar, 50 µm (**b**, height); 100 µm (**b**, length).

within a waveguide to population hopping between waveguides. Initially, consider a system of two degenerate localized quantum states that are coupled to each other with a coupling constant $J_0 > 0$. If the probability amplitude is initially concentrated entirely in one of the states (with a value set to 1), the population in the initially unoccupied state follows $\sin^2(J_0 t)$ as a function of time $t$ and, consequently, for small times, increases as $(J_0 t)^2$. Provided the coupling constant is known, the respective populations of the two states can be understood as a measure of time. We then translate this behaviour to a system of two coupled waveguides. Consider two parallel waveguides, each capable of restricting the motion of particles to one dimension. When these two waveguides are brought in proximity, tunnelling processes allow particle exchange between the waveguides, which can be described by a coupling constant $J_0$ as introduced above. If at a certain point $x = 0$ the entire population is concentrated in one waveguide, then, sufficiently close to $x = 0$, the population of the other waveguide is expected to grow as $(J_0 x/v)^2$ (replacing $t$ with $x/v$). Thus, provided that the coupling constant $J_0$ is known, we can infer the speed $v$ by measuring the spatial population build-up in the initially unoccupied state. For a more formal justification of this measurement scheme, based on modelling the system using coupled Schrödinger equations, we refer to the Methods and ref. 35. We emphasize that this measurement scheme does not determine the direction of motion of particles; thus, it is not a velocity measurement. For more details, see the Methods.

To implement this scheme experimentally, we use a quantum-confined photon gas in a planar optical microcavity[36–39]. In this system, the photons essentially behave like massive particles in two dimensions. More specifically, the mirrors of the microcavity restrict the allowed values

for the photon wavevector $\mathbf{k} = (k_x, k_y, k_z)$ to those that satisfy a resonance condition along the optical axis, that is $k_z = q\pi/D_0$. Here, $q = 1, 2, 3, \dots$ is the longitudinal mode number and $D_0$ is the spatial separation of the mirrors. If the photons travel predominantly in the direction of the optical axis (paraxial limit), the projection of the photon motion onto the resonator plane creates a two-dimensional non-relativistic gas of particles with wavevectors $(k_x, k_y)$. In this projection, the photons acquire an effective mass $m$, whose value follows the energy–mass equivalence $E_0 = m\tilde{c}^2$, where $E_0$ is the energy of the transverse ground mode and $\tilde{c}$ is the speed of light in the resonator medium[36]. The latter follows from the photonic energy–momentum relationship $E_{ph} = \hbar\tilde{c}|\mathbf{k}|$, which, assuming a paraxial limit $|k_{x,y}| \ll |k_z|$, can be approximated by $E_{ph} \approx m\tilde{c}^2 + \hbar^2(k_x^2 + k_y^2)/2m$, with the photon mass $m = \hbar\tilde{c}q\pi/D_0 = E_0/\tilde{c}^2$. In our experiments, we have $m \approx 6.95 \times 10^{-36}$ kg. In this system, the transverse motion of the photons can be precisely controlled by nanostructured mirror surfaces. It can be shown[39] that a local change in the distance between the mirrors $\Delta d(x, y)$ introduces a potential energy $V(x, y)$ for the particles that follows $V(x, y) = -m\tilde{c}^2 \Delta d(x, y)/D_0$. In our experiment, this effect is used to capture photons in one-dimensional waveguide potentials and to couple waveguide pairs in a controlled manner. Owing to a small amount of mirror transmission with a rate of $\gamma \approx 3.7$ GHz, corresponding to a particle lifetime of $\tau = 1/\gamma \approx 270$ ps, particle densities in the resonator plane can be fully reconstructed by imaging the transmitted light onto a camera. This is true regardless of whether the transverse state of the emitted light represents evanescent or propagating solutions, allowing us to study evanescent phenomena at semi-infinitely extended step potentials.

The core of our experimental setup, as described in Fig. 1a, consists of a high-finesse microcavity with a dye medium. One of the mirrors is

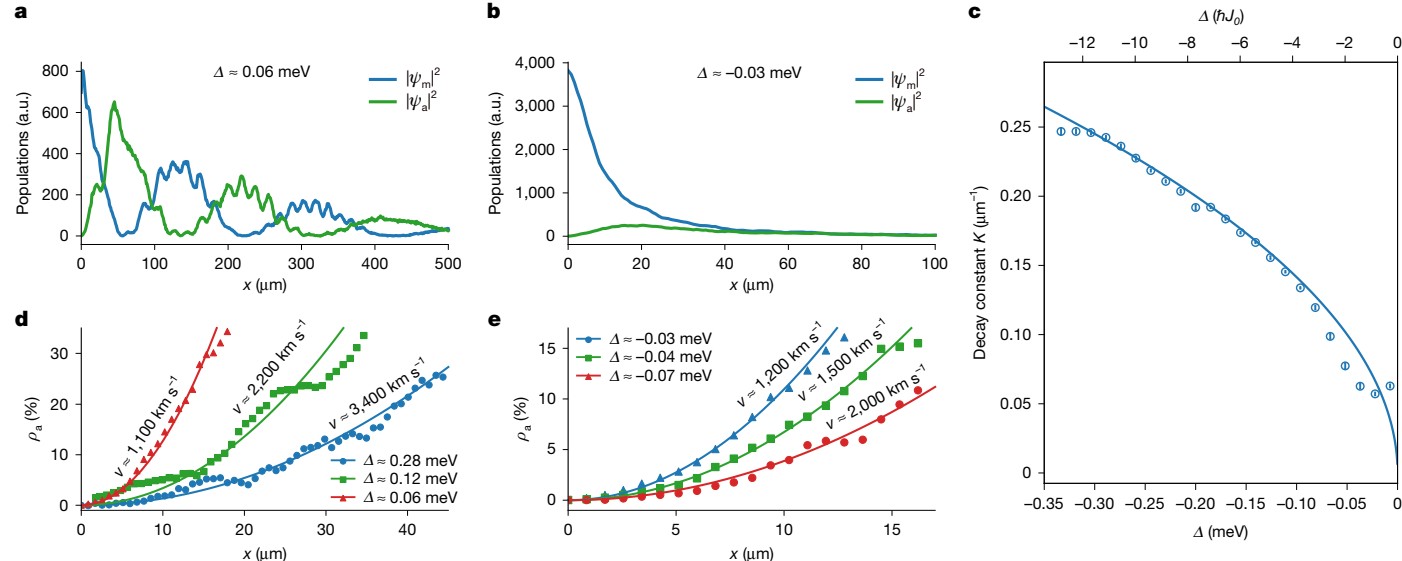

**Fig. 2 | Populations $|\psi_{m,a}|^2$ as a function of energy $\Delta$ and position $x$.**
**a**, Measurement example of the particle densities along the waveguide axis in the main waveguide $|\psi_m|^2$ (blue) and auxiliary waveguide $|\psi_a|^2$ (green) in the classically allowed parameter regime ($\Delta > \hbar J_0$). The fast periodic modulations on top of the population curves stem from a residual reflection at the end of the coupled waveguide system. **b**, Particle densities in the classically forbidden regime. **c**, Decay constant $\kappa$ of the population in the main waveguide for different energies $\Delta$ (see the Methods for further details). The observed decay constant closely follows $\kappa = \sqrt{2m|\Delta|}/\hbar$. Error bars indicate the standard error of the mean.

**d**, Examples of measured relative populations in the auxiliary waveguide $\rho_a = |\psi_a|^2/(|\psi_m|^2 + |\psi_a|^2)$ for classically allowed states. A parabolic fit $\rho_a = (J_0 x/v)^2$ is used to determine the speed of the motion. **e**, Relative populations for evanescent states in which the step is fully reflective. In these states, particles enter the step potential and eventually exit it with their direction of motion reversed. As in the case of the propagating states, a parabolic fit $\rho_a = (J_0 x/v)^2$ is used to determine the speed of the motion. All data shown in this figure, except those in the inset of **c**, are derived from single data-taking events, each comprising 20 optical excitations of the system.

nanostructured to create waveguiding potentials for photons in the transverse plane of the resonator. The main waveguide begins with a linear ramp, allowing us to vary the initial potential energy of the photons by changing the point along the ramp at which we optically pump the system with a non-resonant laser pulse. The pulse duration of 26 ns (full width at half maximum (FWHM)) is considerably longer than all relevant time scales in the system, creating a quasi-stationary scenario for all our experiments. This pumping triggers a lasing process that favours the creation of photons with low kinetic energy at the position of the pump, maximizing the overlap of their wave function with the optical gain. The potential energy of the photons created at a certain position in the ramp is then converted into kinetic energy as the photons move down and reach a region of constant potential. The photons then impinge on the step potential, as shown in Fig. 1b, in which a second waveguide opens up. We refer to this second waveguide as the auxiliary waveguide in the subsequent discussion. Three examples showing the photon densities in the microcavity for different pump spot positions are shown in Fig. 1c. The densities are recorded by imaging light transmitted through one of the mirrors onto a camera. The auxiliary waveguide allows us to determine the particle speed by comparing the translational motion along the waveguide axis with the hopping between waveguides. A prerequisite for this is determining the coupling constant $J_0$ between the waveguides. This can be performed through a wavenumber analysis of observed mode patterns in the classically allowed regime, yielding $J_0 = 2\pi(6.34 \pm 0.01)$ GHz or $\hbar J_0 = (26.22 \pm 0.04)$ µeV. For further details, see the Methods. In the same way, we can also determine all other quantities of interest, such as the height of the step potential $V_0 = (0.538 \pm 0.003)$ meV and the kinetic energy $E$ of the incoming particles.

Figure 2a,b shows the populations $|\psi_{m,a}|^2$ in the main ($\psi_m$) and auxiliary ($\psi_a$) waveguides as a function of position $x$ for specific values of $\Delta = E - V_0 + \hbar J_0$, which corresponds to the kinetic energy of the particles in the step potential. For $\Delta > \hbar J_0$ (Fig. 2a), we observe that the population oscillates between the waveguides as the particles

propagate in the step potential. The observed population decay over several hundred micrometres is because of mirror transmission and can be used to experimentally determine the photon loss rate $\gamma$ introduced before. For $\Delta < -\hbar J_0$ (Fig. 2b), evanescent types of wave functions are observed for which the populations in the waveguides relax to a common value. Figure 2c shows the decay constant of the population in the main waveguide, denoted by $\kappa$, as a function of $\Delta$ (Methods). For sufficiently large $|\Delta|$, the measurement is in excellent agreement with the expected decay constant $\kappa = \sqrt{2m|\Delta|}/\hbar$.

Figure 2d,e shows the relative population in the auxiliary waveguide $\rho_a = |\psi_a|^2/(|\psi_m|^2 + |\psi_a|^2)$ as a function of distance to the potential step for different values of $\Delta$. Calculating the relative populations eliminates common factors affecting both waveguides, such as the exponential decay in the case of negative $\Delta$. In all cases, the data are compatible with a quadratic increase in relative population. By modelling our data with $\rho_a = (J_0 x/v)^2$, we obtain the particle speed $v$ shown in Fig. 3a. It is to be noted that, within the experimental uncertainties, $v$ is a mirror-symmetrical function of $\Delta$. For negative $\Delta$, this means, rather counterintuitively, that the lower the energy of the particles, the faster the particles move within the step potential. We emphasize that the measured speed represents the local speed of particles directly behind the potential step. In the coupled waveguide system, a position-dependent state of motion can arise from an interplay with the coupling energy, even if the potential energy remains constant. However, these effects are not expected when the coupling energy is negligible, specifically when $|\Delta| \gg \hbar J_0$. In this limit, our data align well with $v = \sqrt{2|\Delta|/m}$ for both negative and positive values of $\Delta$ (Fig. 3a). The residual differences are probably caused by a systematic overestimation of the speed in the fitting procedure of up to 6.7% (see the Methods for details). Note that for negative $\Delta$, the speed $v = \sqrt{2|\Delta|/m}$ is proportional to the decay constant $\kappa = \sqrt{2m|\Delta|}/\hbar$, following $v = \hbar\kappa/m$. Introducing the decay length $\lambda = 1/\kappa$, this relation can also be stated as $\lambda = \hbar/mv$, which represents a de Broglie relation for evanescent particle states, connecting the decay length to the speed.

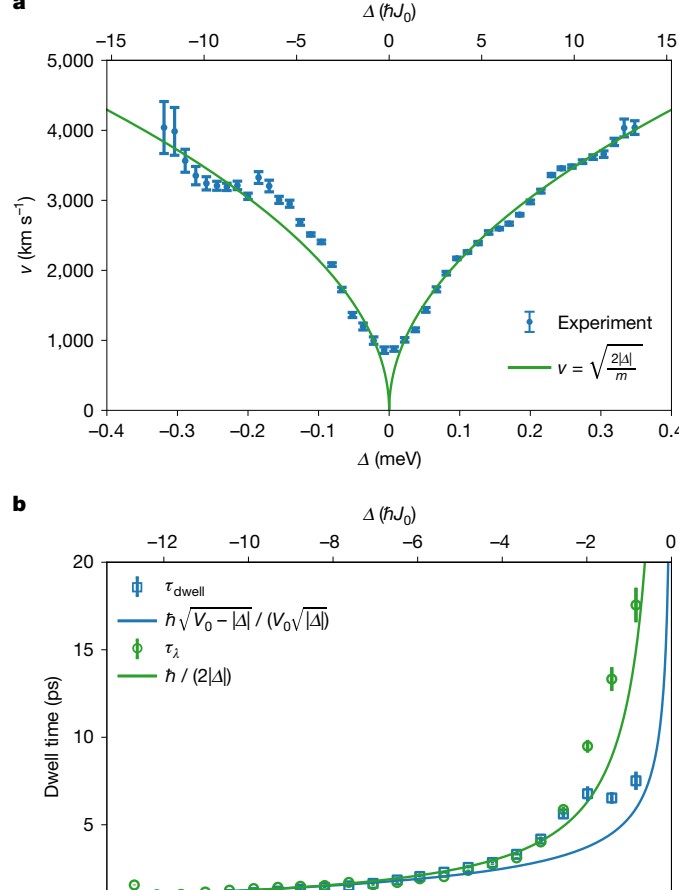

**a**

**b**

**Fig. 3 | Particle speed and dwell time. a**, Particle speed $v$ at the potential step as a function of energy $\Delta$. Blue markers indicate the speed values obtained from the parabolic fitting of the population increase in the auxiliary waveguide. The speed obtained in this way is found to be a mirror-symmetric function of the energy $\Delta$. For sufficiently large $|\Delta|$, the data align well with $v = \sqrt{2|\Delta|/m}$ (green line). For negative $\Delta$, this means that the lower the speed of the incoming particles, the faster the particles move within the step potential. The lowest speed values are found in the vicinity of $\Delta = 0$ and are close to the theoretical prediction[35] $v_{min} = \sqrt{\hbar J_0/m} \approx 777$ km s$^{-1}$ for $J_0 = 2\pi \times 6.34$ GHz. Error bars indicate the standard error of the mean. **b**, Dwell time as a function of energy. The dwell time $\tau_{dwell}$ (squares) describes how long particles are stored in the step potential and can be determined using $\tau_{dwell} = N/j_{in}$, where $N$ is the number of stored particles and $j_{in}$ is the incoming particle current. Based on the energy–speed relation in **a**, a semiclassical estimate of $\tau_{dwell}$ can be introduced using $\tau_\lambda = \lambda/v$, where $\lambda$ is the decay length (circles). This estimate is found to be in good agreement with $\tau_{dwell}$ for a wide range of energies. The two solid lines give the expected behaviour for the dwell time and its semiclassical estimate. The given formulas are derived for a single waveguide step potential. Error bars for $\tau_{dwell}$ indicate the standard error of the mean. Error bars for $\tau_\lambda$ are derived from error propagation.

As an application of the measured energy–speed relation, we consider the dwell time of particles in the step potential[13]. The definition of the dwell time is based on the idea that, in a steady state, the storage time of particles in a reservoir should be given by $\tau_{dwell} = N/j_{in}$, where $N$ is the number of stored particles and $j_{in}$ is the incoming particle current. For a fully reflecting step potential—producing a standing wave pattern in front of the step and an exponentially decaying wave function within it—the incoming and outgoing particle currents can be assumed to have equal magnitudes but opposite directions. In analysing the experiment, the incoming current is taken as $j_{in} = (|\psi_{max}|^2/4)v_{in}$, where $\psi_{max}$

is the maximum amplitude of the wave function in front of the step. The factor of 4 arises from both the standing wave interference effect and the division into incoming and outgoing components. Alternatively, we could use the average density in front of the step along with a division factor of 2. The velocity $v_{in}$ is taken as $v_{in} = \hbar k_0/m$, where $k_0$ is determined from the periodicity of the observed standing wave interference pattern. Note that this determination of $j_{in}$ does not hold in the framework of Bohmian mechanics, as will be discussed below. The dwell time based on this procedure is observed to be within 1–2 ps for a wide range of energies (Fig. 3b). Using the measured energy–speed relation, we can give a semiclassical estimate of $\tau_{dwell}$ by considering $\tau_\lambda = \lambda/v$, where the decay length $\lambda$ corresponds to a typical travel distance into and out of the step potential. The latter is derived from $2\langle \hat{x} \rangle_{x>0} = 2\varrho_0^{-1} \int_0^\infty \varrho(x)x\,dx = \lambda$, where $\varrho(x) = (e^{-x/\lambda})^2$ and $\varrho_0 = \int_0^\infty \varrho(x)dx$. The quantity $\tau_\lambda$ is found to be close to the dwell time for a wide energy range (Fig. 3b). We consider this to be further evidence for the physical significance of the found energy–speed relation.

In further measurements, shown in Fig. 4, we interfere quantum states in the step potential with their spatially mirrored images to detect phase gradients (see the Methods for further details). These phase gradients can then be converted into velocities following equation (1). For $\Delta > 0$ (Fig. 4b, left), we obtain velocities that are compatible with the results derived from the population transfer measurements shown in Fig. 3a. For $\Delta < 0$, the phase gradients are found to be close to zero (Fig. 4b, right). If we translate the residual phase gradients into velocities using equation (1), we obtain on average $v_s = (59 \pm 42)$ km s$^{-1}$, as shown in Fig. 4c, which is consistent with zero. The vanishing of $v_s$ indicates a vanishing net particle current, meaning that the motion becomes non-directional as the step potential becomes fully reflective. The Bohmian interpretation of equation (1) goes further by postulating that $v_s$ is a complete characterization of the state of motion of the particles. Therefore, a vanishing $v_s$ not only indicates non-directionality but also implies that the particles are at rest. Note that this implication is in contradiction with the conclusions drawn from our speed measurements (Fig. 3a).

The assumption that particles in the step potential are at rest also implies a fundamentally different behaviour of other quantities. The dwell time, for example, can be identically defined by $\tau_{dwell} = N/j_{in}$ in both standard quantum mechanics and Bohmian mechanics. As the wave functions are identical, the particle number $N$—defined as the integral of the modulus of the wave function squared within the step potential—is the same in both cases. However, the incoming particle current $j_{in}$ cannot, in general, be assumed to be identical. Scattering at a fully reflective potential step—when the energy of the particle is lower than the step potential—gives rise to eigenfunctions that can be chosen to be purely real-valued. The corresponding absence of phase gradients indicates, in the Bohmian framework, that the particles are at rest everywhere in the system. In other words, there are no particle currents—specifically, no incoming current into the step potential. Consequently, the ratio $N/j_{in}$ diverges, implying an infinite dwell time. Note that $j_{in} = 0$ is not merely a consequence but also a requirement within Bohmian mechanics to remain consistent with the guiding equation inside the potential step, which also predicts the absence of motion—and thus an infinite dwell time. Thus, the dwell time for scattering at a semi-infinite step potential is an example of a physical quantity that, although identically defined and measurable in both Bohmian mechanics and standard quantum mechanics, yields different values in the two theories.

Furthermore, an absence of particle motion in the step potential—and consequently an infinite dwell time—implies that losses are essential to the scattering process observed in our experiment, as every particle entering the potential step would eventually be lost because of the dwell time exceeding the finite lifetime of the particles of 270 ps. By contrast, the wave function observed in the vicinity of the potential step closely matches the wave function expected for a lossless system (see, for example, Fig. 2c), thereby implying that losses are not significant in

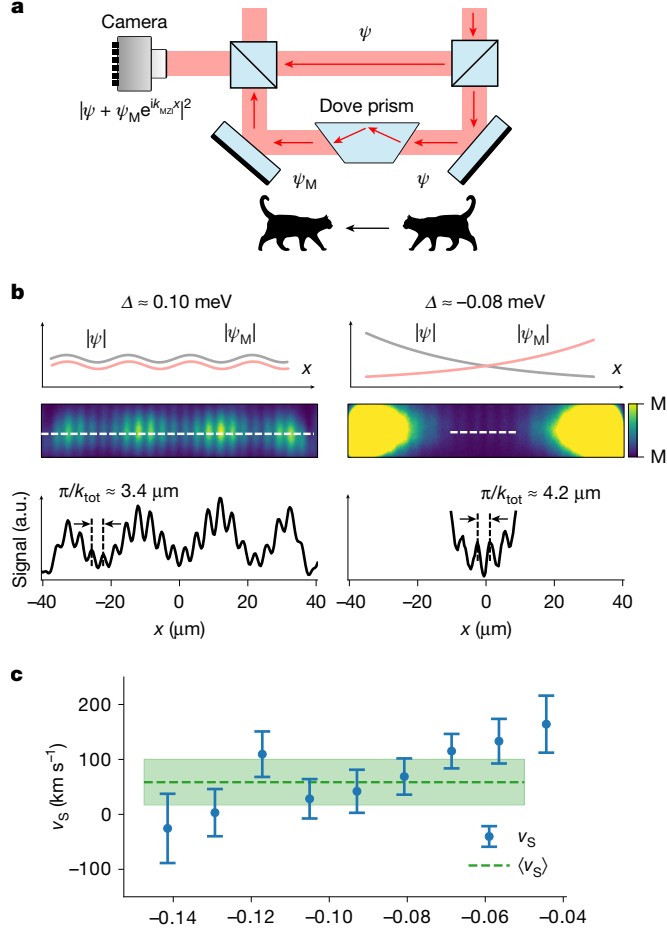

**Fig. 4 | Phase gradient (Bohmian) velocity $v_S$. a**, To measure the phase gradient of quantum states in the step potential, we incorporate a Mach–Zehnder interferometer into our experimental setup. A Dove prism is placed in one arm of the interferometer to create a mirrored image of the input state. The wavenumber associated with the observed interference pattern at the camera, $k_{tot} = k_{MZI} + k_S$, has two contributions: a component related to the alignment of the interferometer, $k_{MZI}$, which can be determined in an independent measurement, and a component associated with the intrinsic phase gradient of the quantum state $k_S$. **b**, The top row shows sketches of wave functions and their mirror images, $\psi$ and $\psi_M$, interfering at the output of the interferometer in the classically allowed (left) and forbidden (right) regimes, respectively. The middle row displays images of experimentally obtained interference patterns. The bottom row shows the corresponding line-integrated densities. From the periodicity of the obtained interference patterns, we extract $k_{tot}$. This is then converted to the velocity $v_S = \hbar k_S/m = \hbar(k_{tot} - k_{MZI})/m$. For the two shown cases, we obtain $v_S \approx 2{,}400$ km s⁻¹ for $\Delta \approx 0.1$ meV and $v_S \approx (31 \pm 42)$ km s⁻¹ for $\Delta \approx -0.08$ meV. **c**, Blue markers indicate the velocity $v_S$ in the classically forbidden regime, derived using the procedure described above. The green dashed line shows the average value, $\langle v_S \rangle = (59 \pm 42)$ km s⁻¹, of all measurements with $\Delta \le -0.05$ meV. Error bars indicate the standard error of the mean.

the scattering process. Thus, the wave picture based on the Schrödinger equation and the particle picture based on the guiding equation offer inconsistent accounts of the significance of losses in the scattering process, within the Bohmian interpretation of our experiment. Even independently of the speed measurements presented in this work, this discrepancy casts doubt on whether the guiding equation accurately captures the temporal dynamics of the scattering process.

In this work, we have measured an energy–speed relationship for tunnelling particles based on the analysis of two orthogonal motions in a system of two coupled waveguides: translational motion along the

waveguides and population hopping between the waveguides. Quantitatively, we see agreement with an energy–speed relation $v = \sqrt{2|\Delta|/m}$. For $\Delta < 0$, this implies that particles move faster the less kinetic energy they have, that is, the more negative their local kinetic energy becomes. If the speed $v$ is translated into the time $\tau_\lambda = \lambda/v$, for $E < V_0$ and the decay length $\lambda$, this accurately reproduces the experimentally observed dwell time of the particles over a wide energy range. If $v$ is translated into the time $\tau_b = b/v$, for $E < V_0$ and the barrier width $b$, the result is the tunnelling time proposed by Büttiker and Landauer[12,13,40]. These straightforward applications of the measured energy–speed relation to evanescent phenomena suggest that this relation plays a fundamental part in quantum tunnelling. More broadly, our work indicates that, in addition to phase gradients, amplitude decays in quantum mechanical wave functions serve as indicators of motion, following a de Broglie relation $\lambda = \hbar/mv$, where $\lambda$ is the decay length and $v$ denotes the nondirectional particle speed measured in this work. This contrasts with Bohmian mechanics, which considers phase gradients as the sole indicators of motion.

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

# Methods

## Experimental setup

Our experiments are performed in a high-finesse optical microcavity consisting of two dielectric mirrors. We use Rhodamine 101 dissolved in ethylene glycol as an active optical medium at a concentration of 1 mmol l$^{-1}$. One of the mirrors is fixed, whereas the other is connected to three piezo-electrical actuators, enabling precise adjustments to the cavity length and tilt. The mirror separation is set approximately to $D_0 = 15$ μm. Both the mirror separation and the angular orientation of the two mirrors are stabilized during the experiment. For the mirror separation, we measure the wavelength of the emitted light and provide electrical feedback to the piezos to stabilize the desired cavity length. For the angular alignment, a new technique based on the inverse solution of the Schrödinger equation was developed[41]. We use non-resonant pumping at a wavelength of 532 nm to create regions of high chemical potential, in which photons condense[36,37] and form coherent states of light at a wavelength of about 650 nm. The laser pulse duration is around 26 ns (FWHM) at a pulse repetition rate of 500 Hz. A spatial light modulator is used to move the pumping spot in the plane of the cavity, allowing different particle energies to be probed. We use a band-pass filter with a central wavelength of 650 nm and a width of 10 nm (FWHM) before the cameras to isolate the signal from the background. We furthermore use a D-mirror to split the light emitted from the cavity and to image different spatial regions of the signal onto different cameras. This approach allows us to isolate high signals in the waveguide before the potential step at $x = 0$ from (potentially) low signals in the coupled waveguide region.

## Sample preparation

We apply a new nano-structuring method[42] to create waveguiding potentials for the photons in the transverse plane of the resonator. The main waveguide (Fig. 1b) begins with a linear ramp, allowing us to vary the initial potential energy of the photons by changing the point along the ramp at which we optically pump the system. The potential energy of the photons is then converted into kinetic energy as the photons move downward and reach a region of constant potential. The photons encounter the step potential in which a second waveguide opens up. The coupled waveguide potentials are designed to gradually diminish at large distances to minimize back reflections from the end. The height map of the mirror surface (Fig. 1b) is determined through Mirau interferometry.

## Measurement scheme for determining particle speeds

For a formal justification of the experimental method, we model the coupled waveguide system using the following set of steady-state Schrödinger equations (valid in a quasi-continuous-wave configuration):

$$E\psi_m = -\frac{\hbar}{2m}\frac{d^2\psi_m}{dx^2} + V_0\psi_m + \hbar J_0(\psi_a - \psi_m), \qquad (2)$$

$$E\psi_a = -\frac{\hbar}{2m}\frac{d^2\psi_a}{dx^2} + V_0\psi_a + \hbar J_0(\psi_m - \psi_a), \qquad (3)$$

where $\psi_m$ represents the wave functions in the main waveguide, $\psi_a$ represents the wave functions in the auxiliary waveguide, $E$ is the total energy, $V_0$ is the potential energy and $J_0$ is the coupling constant between the waveguides. An ansatz for the solution of these equations can be written as

$$\psi_m \propto \cos(k_1 x)e^{ik_2 x}, \qquad (4)$$

$$\psi_a \propto \sin(k_1 x)e^{ik_2 x}. \qquad (5)$$

This ansatz describes the motion of particles as plane waves, $\exp(ik_2 x)$, which, in case where $E < V_0$, transitions into an evanescent solution. The relative population of the waveguides is determined by the population transfer factors $\cos(k_1 x)$ and $\sin(k_1 x)$. Using the ansatz provided above, the increase in population in the auxiliary waveguide can be shown to follow

$$\rho_a(x) = \frac{|\psi_a|^2}{|\psi_m|^2 + |\psi_a|^2} \approx |k_1|^2 x^2, \qquad (6)$$

for small $x$. Thus, a parabolic increase in the relative population in the auxiliary waveguide is expected, regardless of the type of motion. The coefficient $|k_1|^2$ is experimentally accessible. By applying the ansatz, equations (4) and (5), to the coupled Schrödinger equations, we find that $k_1$ and $k_2$ satisfy the relation

$$k_1 = \frac{mJ_0}{\hbar k_2}. \qquad (7)$$

This relation holds irrespective of whether the solution describes a propagating or evanescent state. It forms the basis of our measurement scheme, as it connects the motional state ($k_2$) with the relative population in the waveguides ($k_1$), our measurement quantity. To justify the method, it is not strictly necessary to use the full information provided by equation (7). The key point of interest is that $k_1$ is proportional to $J_0$ (given that $k_2$ is independent of $J_0$, which holds true for $|E - V_0| \gg \hbar J_0$; ref. 35). This relationship is, in principle, already evident from the coupling between two localized states, as discussed in this study. As $J_0$ is a property of our measurement device and not of the state of motion, we factor it out as a pre-factor, $|k_1|^2 = (J_0/v)^2$, resulting in

$$\rho_a(x) \approx |k_1|^2 x^2 = \left(\frac{J_0}{v}\right)^2 x^2. \qquad (8)$$

The remaining factor $v$ has the dimensions of a velocity and an undetermined sign. Most importantly, this quantity determines the spatial length scale of the population build-up in the coupled waveguide system—an effect that genuinely represents a spatio-temporal phenomenon, distributing particles in space ($x$) relative to a temporal reference ($J_0$). We regard this as a property that can be attributed only to a form of motion. It is important to emphasize that this measurement scheme does not determine a sign of $v$. A sign implies directionality, indicating net particle transport. Any measurement scheme sensitive to direction must necessarily yield a velocity of zero for an evanescent state of the form $\exp(-x/\lambda)$, as these states do not imply net particle transport. Conversely, any scheme that assigns a non-zero speed value to a state of this form, which is true for the scheme presented above, must be insensitive to direction.

## Statistical error analysis

All errors shown in the figures represent standard errors of the mean. For Fig. 3, these were obtained by averaging 156 independent measurements performed over several days. For Fig. 4, we average over 19 measurements. No dataset was excluded. Apart from statistical errors, we expect systematic errors mainly from two sources. This will be discussed below.

## Determination of the energy $E$ of the incident particles

The (kinetic) energy $E$ of the incoming particles is obtained by analysing the standing wave pattern resulting from the superposition of the incident particle stream with the particle stream reflected from the potential step. First, the observed intensity pattern is transformed to Fourier space. Then, the dominant spatial frequency $k_{max}$ is extracted. This corresponds to a wavenumber of the particles of $k_0 = k_{max}/2$. The additional factor of 2 arises because the Fourier analysis is based on

intensity patterns rather than the complex amplitude of the incident wave. With this, the energy of the incident particles follows as $E = (\hbar k_0)^2/2m$.

## Determination of the coupling constant $J_0$

For determining $J_0$, we use the relation $J_0 = \hbar k_1 k_2/m$ (ref. 35). This relation applies to classically allowed motion in the coupled waveguide system (and remains valid in the classically forbidden regime). The wavenumber $k_1$ represents the spatial periodicity of the population transfer between the main and auxiliary waveguides. This is derived by fitting the relative population of the waveguides with harmonic functions, for example, $\rho_a = \sin^2(k_1 x)$. The wavenumber $k_2$ describes the motion of the particles within a waveguide of the coupled waveguide system. It is determined similarly to $k_0$ by analysing a standing wave pattern created by a small residual back reflection from the open end of the coupled waveguide system. In this way, we obtain $J_0 = 2\pi(6.34 \pm 0.01)$ GHz for the coupling constant when averaging over all 156 measurements.

## Determination of the decay constant $\kappa$

For $\Delta < 0$, we derive a position-dependent decay constant in the main waveguide using $\kappa(x) = (-1/2) \times d[\ln(|\psi_m(x)|^2)]/dx$. The decay constant $\kappa$ is considered to be the maximum value encountered in $\kappa(x)$, which in all observed cases is located close to the potential step.

## Determination of the step height $V_0$

For determining $V_0$, we use the relation $V_0 = E - (\hbar k_2)^2/2m + \hbar J_0$, which is valid asymptotically in both the classically allowed and classically forbidden regimes[35]. In the classically allowed regime, the wavenumber $k_2$ describes the phase gradient of the quantum state within a waveguide of the coupled waveguide system. In the classically forbidden regime, $k_2$ becomes imaginary and describes the decay of the quantum state, $k_2 = i\kappa$. Consequently, $(\hbar k_2)^2/2m$ becomes negative. To a good approximation, both regimes yield agreeing values for $V_0$. When averaging over all 156 measurements, we obtain $V_0 = (0.538 \pm 0.003)$ meV.

## Determination of the speed $v$ and its expected systematic errors

The particle speed given in Fig. 3a is obtained by fitting the measured relative population in the auxiliary waveguide, $\rho_a(x)$, to the model $f(x) = [J_0(x - x_0)/v']^2$. The fitting procedure aims to find the value for $v'$ that minimizes the error $R(v') = \int_{x_0}^{x_{max}} dx[\rho_a(x) - f(x)]^2$. As the coupling constant $J_0$ can be independently determined, the fitting procedure depends on two input parameters: $x_0$ and $x_{max}$, which are the lower and upper bounds of the error integral, respectively. These parameters are considered the main sources of systematic errors in our data evaluation. Ideally, $x_0$ is simply defined by the position of the potential step; however, owing to the limitations of the nano-structuring method used for sample preparation, which has a minimal structure size of approximately 4 μm (FWHM of a point-like structure)[42], the potential step is not perfectly sharp. In our data analysis, we treat $x_0$ as a global parameter set identically across all measurements. Its uncertainty is assumed to be 2 μm. Varying $x_0$ by this amount systematically shifts the obtained speed values by an average of 8.5% compared with the results presented in Fig. 3a. The second parameter, $x_{max}$, determines the extent of data considered in the fitting procedure. Generally, using more data will enhance the reliability of the speed determination. By contrast, if $x_{max}$ is set too high, the accuracy of the parabolic approximation $f(x)$ decreases, introducing systematic errors. This systematic error can be estimated as follows. First, we consider the case of classically allowed motion in which the experimentally obtained data are expected to be of the form $\rho_a(x) = \sin^2(J_0 x/v) + \eta(x)$. Here, $\eta(x)$ is additive noise with an expectation value of zero, $\langle \eta(x) \rangle = 0$. For an individual measurement outcome, the optimal value for $v'$ can be determined by solving $dR(v')/dv' = 0$ for $v'$. This condition can be expressed as

$\int_{x_0}^{x_{max}} dx 2(df/dv')(f - \rho_a) = 0$. Averaging over many realizations, $\rho_a$ is replaced by $\langle \rho_a \rangle = \sin^2(J_0 x/v)$ due to the assumed properties of the noise. The (average) optimal value for $v'$ is then obtained from the equation $\int_{x_0}^{x_{max}} dx 2(-2J_0 x^2/v'^3)[(J_0 x/v')^2 - \sin^2(J_0 x/v)] = 0$. Solving this equation analytically yields a cumbersome expression for $v'$, which we will not reproduce here. For this work, it is sufficient to approximate the solution by a series expansion, giving $v'/v - 1 = (5/42)\varepsilon + (1543/31752)\varepsilon^2$. Here, $\varepsilon$ is the maximum relative population considered in the fitting procedure, which is related to $x_{max}$ by $x_{max} = (v/J_0)\arcsin\sqrt{\varepsilon}$. In our analysis of the data in the classically allowed regime, we choose $\varepsilon = 0.25$, which yields $v'/v - 1 = 0.033$. This indicates that using data up to a relative population of 25% in the auxiliary waveguide introduces a systematic error in the extracted speeds of approximately 3% because of the parabolic approximation. This is comparable to the statistical uncertainties in our measurements. A similar calculation can be performed in the classically forbidden regime, in which we assume the experimentally obtained data to be of the form $\rho_a(x) = \sinh^2(J_0 x/v)/[\sinh^2(J_0 x/v) + \cosh^2(J_0 x/v)] + \eta(x)$. The result of this calculation is $v'/v - 1 = (25/42)\varepsilon + (24463/31752)\varepsilon^2$. Thus, for the same value of $\varepsilon$, the systematic deviation for $v'$ due to the parabolic approximation turns out to be five times larger in the classically forbidden regime than in the classically allowed regime. This needs to be considered when choosing an appropriate value for $\varepsilon$. Furthermore, it must be considered that data up to a specific relative population are not necessarily available for all energies $\Delta$. In the classically forbidden regime, the point at which a certain population level is reached progressively moves deeper into the step potential as $\Delta$ becomes smaller and the particle speed increases. Moreover, the decay constant increases, causing the signal to decay more rapidly. This can prevent observing specific population levels due to noise. Therefore, in our data analysis, we adjust $\varepsilon$ as we move deeper into the classically forbidden region. More specifically, we choose $\varepsilon = 0.1$ for $-0.2$ meV $< \Delta < 0$ and $\varepsilon = 0.05$ for $\Delta < -0.2$ meV. The expected systematic deviations for the determined speed due to the parabolic approximation are 6.7% and 3.2%, respectively.

## Determination of the velocity $v_s$

We perform interferometric measurements in which we interfere the wave function with a spatially mirrored image of itself to detect phase gradients. For this purpose, we use a Mach–Zehnder interferometer, equipped with a Dove prism in one of the arms to mirror the image in one direction. The determination of the total wavevector, $k_{tot}$, which describes the overall periodicity of the observed interference pattern, is performed like the one described above, namely, by analysing the Fourier spectrum of the images. Crucially, the observed interference patterns—and thus the wavevector $k_{tot}$—depend not only on the intrinsic phase gradients of the examined wave functions but also on the alignment of the interferometer. To determine the contribution of the interferometer, we investigate reference modes created in additional surface structures independent from the coupled waveguide system. These reference modes are designed to have a constant intrinsic phase. Their analysis yields the wavevector contribution due to the alignment of the interferometer, denoted as $k_{MZI}$. This contribution is subtracted from $k_{tot}$ to determine the intrinsic phase gradients of the wave functions and the associated velocity $v_S = \hbar(k_{tot} - k_{MZI})/m$.

## Data availability

The data supporting this study are available at https://doi.org/10.6084/m9.figshare.28847645. Source data are provided with this paper.

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

**Acknowledgements** This work received funding from the European Research Council under the Horizon 2020 research and innovation programme of the European Union (grant agreement no. 101001512) and from the Dutch Research Council (grant no. OCENW. KLEIN.453).

**Author contributions** V.S. and M.P. had equal roles in carrying out the experiment; V.S., M.P. and J.K. performed the data analysis; C.M. and C.T. contributed to the experimental methods; J.K. initiated and supervised the project; all authors contributed to the interpretation of the data and the drafting of the paper.

**Competing interests** The authors declare no competing interests.

**Additional information**
**Correspondence and requests for materials** should be addressed to Jan Klaers.
