## [Peer Review File · Nature]

Energy-speed relationship of quantum particles challenges Bohmian mechanics

Corresponding Author: Dr Jan Klaers

Version 1:

Reviewer comments:

Referee #1

(Remarks to the Author)

The paper proposes and describes an experiment to measure the velocity distribution as a function of energy for quantum systems in what the dub 'classical forbidden motion.' The authors claim that the empirical distributions pose a challenge to Bohmian mechanics.

The article can be more clearly written. For instance, the conclusion of the paper is not clear from the abstract, or state in the introduction; it is merely mentioned 'in passing' at the end. Since this is the 'Big Deal' of the paper, it should be emphasized right away.

The language is often not very precise. This can be seen immediately from the title: "Measuring the speed of classically forbidden motion." The property of having a velocity is a property of systems, and even if it is clear what the authors mean, it is weird to say that motion has speed. Perhaps one could just write: "Speed measurements for states of classically forbidden states of motion." This is a very minor point, but since it is not difficult to fix floppy language, I see no reason not to do it. For instance, throughout the paper the authors write as if there are particles in quantum mechanics. This is, strictly speaking, not true, as the authors know, because the quantum state is the wave function. However, physicists, including the authors, still talk about "quantum particles" because they think that this particle-talk is effective, and so they use it. However, to avoid easy objections and misunderstandings, I would suggest the authors to modify sentences like this: "quantum mechanics describes the motion of particles using wave functions" (abstract) with something more cautious like this: "quantum mechanics describes the motion of systems using wave functions."

The discussion section, where the implications of the results are supposed to be presented, is too short and cryptic. If the paper deserves publication, it is because it has something interesting to say about Bohmian mechanics. Indeed, the argument presented seems to be like this:

- 1- Bohmian mechanics has particles, evolving in a particular way;
- 2- The Bohmian predictions (about v distribution in classically forbidden states) are different from the 'standard' quantum predictions
- 3- Experiments show Bohmian predictions to be false (or problematic, or whatever).

First, it would have been extremely helpful to have something similar to this argument written in the paper to make explicit the role and significance of the experiment discussed.

Second, premise 2 has been disputed in the literature. Indeed Bohmians (at least most of them) are adamant that Bohmian mechanics and quantum mechanics make the very same predictions (even in principle); see here:

<https://link.springer.com/article/10.1007/BF01049004> (preprint here: <https://arxiv.org/abs/quant-ph/0308039>).

So, if Duerr Goldstein and Zanghi are correct, premise 2 is false, and the authors' argument is unsound. I am not saying that the authors are necessarily wrong in thinking that this creates a puzzle for the Bohmians; but they need at least to recognize that their argument relies on a premise P2 which the Bohmians reject. This is a big deal and it's an ongoing debate (see, for related debates, e.g.

<https://www.nature.com/articles/s41598-018-38261-4>

<https://www.mdpi.com/2073-8994/16/10/1325>

<https://www.nature.com/articles/s41598-024-53777-8>

<https://link.springer.com/article/10.1007/s10701-024-00798-y>

Without this discussion, I cannot recommend this paper for publication.

Referee #2

(Remarks to the Author)

The authors present a study in which a photonic microcavity is set up in a configuration such that a double-well potential emerges in which population oscillations can be observed via evanescent coupling through a tunnel barrier with controllable height and width. The observed dynamics can be related back to tunnelling and dwell times, and the authors find that under certain assumptions dwell times conform to those proposed by Büttiger/Landauer which have since been empirically confirmed. Conclusions are drawn in regards to predictions for tunnelling dynamics in the Bohmian interpretation.

This is a beautiful experiment with very compelling data. There is a strong case for the system being a valuable contribution to the ongoing discussion of tunnelling dynamics.

The link to Bohmian mechanics is, in my view, a distraction. Bohmian mechanics is notoriously adaptable to whatever is thrown at it, and this particular aspect of Bohmian mechanics is not particularly well developed. The main discussion that I was able to dig up on the hydrodynamics of Bohmian trajectories [Bittner et al., The Journal of Chemical Physics 112 (2000)] in tunnelling scenarios is not referenced here, although clearly that would have been very relevant.

On balance, I'm not convinced that this work has enough novelty and impact for publication in Nature, it would be more suitable for consideration in Nature Physics. I'd be happy to reconsider depending on what my fellow reviewers and the editor have to say.

Referee #3

(Remarks to the Author)

This paper uses a novel approach to imprint both a step potential and a transverse double waveguide structure onto a photon gas excited in a microcavity. Using the tunneling coupling between the waveguides as a clock, the authors then probe the time spent in the forbidden region after a step, as a function of energy deficit, confirming the symmetric predictions of the Büttiker-Landauer time, aside from a small correction they interpret as due to the back-action of the measurement. They discuss the relation of the observed energy-velocity relationship to the Bohm picture of quantum mechanics.

This is a highly original and impressive experiment, and I believe the observations can contribute to the ongoing elucidation of the tunneling-time problem. The methodology appears sound and the paper places itself in the appropriate context, citing the essential prior work so far as I can tell.

There is one central point which gives me pause, and although I'm prepared to give the authors the benefit of the doubt that the current treatment is correct, I believe that before publication this point needs to be more rigorously supported in the text so that readers like myself feel confident in the reasoning rather than merely trusting the authors. This is the discussion around lines 69-73 about the dependence on the build-up of population in a second waveguide, and its dependence on the speed. I would expect that for two wells or waveguides coupled with a coupling constant J_0 , and an initial amplitude of A_1 in well 1 and 0 in well 2, the amplitude to be in well 2 would initially grow as a function of time as $J_0 \cdot A_1 \cdot t$, independent of the velocity. I would ask the authors to explain in much more detail the origin of their $J_0 \cdot x/v$ prediction, and in particular whether it is valid both in cw and pulsed configurations, or only in one.

My impression is that they assume the coupling begins abruptly at $x=0$, and that (at all instants t), the population in waveguide 2 at a given position x involves the coherent sum of population already present in waveguide 2 at $x-\delta_x$ and the population present in waveguide 1 at x , such that the effective coupling time is x/v . On the one hand, this seems plausible, but especially given their desire to draw conclusions about the average v^2 in cases where the phase is constant and the Bohm description involves no motion at all, I would have liked to hear them comment on the dependence on the sign of v . They give a formula for the probability, not the amplitude, which depends only on v^2 ; but if there is a superposition of backward- and forward-propagating waves, does the intuitive argument about interference in the forward direction actually go through? Or have I misunderstood their argument from the start? Is it even clear that this is the relevant timescale, and that the buildup is not determined by the duration of the entire pulse, for instance?

If the authors can provide a satisfactorily clear and (semi-)rigorous justification for the interpretation of their coupling data in terms of a velocity, then I would recommend this paper for publication.

Version 2:

Reviewer comments:

Referee #1

(Remarks to the Author)

The authors have considerably improved the manuscript.

However, they take more time in responding to my criticisms (especially empirical equivalence with Bohmian Mechanics) in the "response to reviewer" letter than in the paper. From the manuscript alone, I would not know where they are coming from and where they want to go regarding the claim they make about Bohmian mechanics.

(Even if they could have explained their position better. In fact, in the response letter they write: "we do not believe that agreement in the density is sufficient to justify the broader claim that the two theories make identical predictions." But why

not? What would they need more than that? Isn't the only thing (probability density) that quantum mechanics provide to connect the formalism with the experiments? They would need to explain better to convince anyone of this). So, the response to this criticism of mine (i.e. the connection with Bohmian mechanics is too quick) in the paper is insufficient. One possibility is to insert what they wrote in the response letter (which is instead more satisfactory) in the paper. However, this might destroy the unity of the article as is now, so I am not positive it is a good idea. Another possibility is to briefly mention the possibility of using this experiment to 'test' Bohmian mechanics without entering into too many details and do that in a future work. I would be unhappy with that, because I do not like claims which are unsubstantiated. Also, the experiment is nice, but it may be difficult to defend a paper about the experiment alone without the connection with Bohmian mechanics. Honestly, I am not sure about that the best course of action would be, and it presumably should be left to the editor.

Referee #2

(Remarks to the Author)

In my initial review I expressed that I wasn't a big fan of the confusing connection the authors were trying to make to Bohmian mechanics. With the discussion in the reply prompted by Referee 1, and the subsequent changes, this aspect of the paper has now become much clearer. I also agree that the tension between "orthodox" interpretations and the Bohmian predictions for tunnelling dynamics was already well covered in some of the references provided, although I would even in this new version suggest that this could be more clearly stated.

The bottom line is thus that the results seem to be a clear refutation of the Bohmian prediction for dwell time, which with this new version I can agree is big news. I am convinced that this will be picked apart from every possible direction by both the orthodox and the Bohmian community, but at least it adds to the so far very scarce body of empirical evidence that allows this discussion to take place at all.

Having said that, there is one thing that I would like to see the authors address before this can be published. In the article by Grace Fields (ref 6 in the new manuscript), the author discusses the entire program of using dwell times for vindicating (or ruling out) the Bohmian interpretation. They conclude that this idea was ultimately abandoned by its original proponents, and I allow myself to use a direct quote from [Fields, 2022]:

Bedard (1997), for example—in a direct response to Cushing—argues that even if his experimental proposal could measure de Broglie-Bohm tunnelling time, its outcome would not be able to falsify de Broglie-Bohm theory without simultaneously falsifying the standard interpretation. This is because, she argues, the two interpretations “(for all practical purposes) make the same predictions”; they only differ in their interpretation of experimental results (Bedard 1997, 186). Thus, although quantum tunnelling time in the standard interpretation and quantum tunnelling time in de Broglie-Bohm theory are “two different properties which are not coextensive and are perhaps measurable in different ways” (186), an experiment designed to measure tunnelling time in one is still a well-defined experiment in the other. If it produces an unexpected outcome, it has provided a falsification of both interpretations.

Belousek (2005, 680) seems to agree:

“Regarding the question of whether the ‘transit’ or ‘tunneling’ times in Bohmian mechanics constitute excess empirical content over ‘orthodox’ quantum mechanics (cf. Ref. 31), I am of the view that, while the ontology of particles following definite trajectories does constitute surplus physical content, this does not generate any excess empirical content in the sense of novel predictions. Instead of novel prediction, Bohmian mechanics allows a more detailed interpretation, and perhaps a more satisfactory explanation, of the measurement outcomes of certain experiments in terms of the dynamical quantities definable within its own theoretical framework. What one has here is a case not of excess empirical content but rather of the well-known ‘theory-ladenness of observation’.”

Cushing, in a response published as a postscript to Bedard's paper, agrees that his original proposal is unviable, but for a slightly different reason. The problem, in his view, lies not only with how to interpret a successful observation of de Broglie-Bohm tunnelling time, but with whether de Broglie-Bohm tunnelling time is observable at all (Bedard 1997, 186).

I suggest, based on the conclusions of Section 4, that there is a simple explanation for why we should not expect to be able to construct a crucial test using tunnelling time. The possibility of a tunnelling-based experimental test of de Broglie-Bohm theory is in principle precluded by exactly the same feature that preserves its empirical equivalence with the orthodox interpretation in other contexts: namely, the fact that we only have access to the indeterministic behaviour of the wavefunction, not the underlying deterministic dynamics (see Section 4.2).

Now, the authors of the manuscript we're considering are perhaps purposefully avoiding to wade into this discussion in any kind of depth, with the mere statement that their observations do not conform to [Bohmian] guiding-wave predictions. I think it would be worth to extend the discussion properly to capture some of the already existing debate on the literature on whether something can be learned at all from this kind of experiment.

Response to Referees – January 16, 2025

Manuscript: Sharoglazova et al., Measuring speed in classically forbidden states of motion, 2024-05-10504A-Z

We are grateful to the Referees for supporting this review process and allowing us to explain and, where necessary, refine our arguments through their assessments. We hope that we have implemented the suggested points in a way that has led to an overall improved manuscript.

In the revised manuscript, the main changes are:

- We have rewritten the discussion on Bohmian mechanics and the implications of our experiment, including mention of our main result in the abstract.
- We have added a more formal justification of our measurement scheme in the Methods section.
- We have added references to relevant literature on tunnelling and arrival times, particularly from the Bohmian perspective.

What follows is a point-by-point response to the Referees' comments.

Referee #1

R1.1 The article can be more clearly written. For instance, the conclusion of the paper is not clear from the abstract, or state in the introduction; it is merely mentioned 'in passing' at the end. Since this is the 'Big Deal' of the paper, it should be emphasized right away.

Following the Referee's suggestions, we have expanded the discussion on Bohmian mechanics and the implications of our experiment. This includes presenting explicitly the main conclusion in the abstract.

R1.2 The language is often not very precise. This can be seen immediately from the title: "Measuring the speed of classically forbidden motion." The property of having a velocity is a property of systems, and even if it is clear what the authors mean, it is weird to say that motion has speed. Perhaps one could just write: "Speed measurements for states of classically forbidden states of motion."

In our view, 'speed' can be considered a property of motion, much like 'directionality.' To avoid potential misunderstandings, however, we have reworded the title to: 'Measuring speed in classically forbidden states of motion.'

R1.3 This is a very minor point, but since it is not difficult to fix floppy language, I see no reason not to do it. For instance, throughout the paper the authors write as if there are particles in quantum mechanics. This is, strictly speaking, not true, as the authors know, because the quantum state is the wave function. However, physicists, including the authors, still talk about "quantum particles" because they think that this particle-talk is

effective, and so they use it. However, to avoid easy objections and misunderstandings, I would suggest the authors to modify sentences like this : “quantum mechanics describes the motion of particles using wave functions” (abstract) with something more cautious like this:” quantum mechanics describes the motion of systems using wave functions.”

We understand the point raised by the Referee; however, we do not fully agree. In our view, the example mentioned in the abstract by the Referee is not the result of imprecise language. Rather, the deeper reason for this type of language lies in the concept of wave-particle duality, which is intrinsic to the canonical interpretation of quantum mechanics.

We agree that this often results in statements that may appear to lack the level of precision sought by the Referee (and others). However, we consider such statements appropriate within the canonical framework of quantum mechanics. The assumption that quantum mechanics is solely about wavefunctions and does not involve particles, as suggested by the Referee, implies an ontological clarity that, in our understanding, canonical quantum mechanics does not provide. For this reason, we do not believe this constitutes an alternative that would reduce potential misunderstandings.

R1.4 The discussion section, where the implications of the results are supposed to be presented, is too short and cryptic. If the paper deserves publication, it is because it has something interesting to say about Bohmian mechanics.

Indeed, the argument presented seems to be like this:

- 1- Bohmian mechanics has particles, evolving in a particular way;
- 2- The Bohmian predictions (about v distribution in classically forbidden states) are different from the ‘standard’ quantum predictions
- 3- Experiments show Bohmian predictions to be false (or problematic, or whatever).

First, it would have been extremely helpful to have something similar to this argument written in the paper to make explicit the role and significance of the experiment discussed. Second, premise 2 has been disputed in the literature. Indeed Bohmians (at least most of them) are adamant that Bohmian mechanics and quantum mechanics make the very same predictions (even in principle); see here: <https://link.springer.com/article/10.1007/BF01049004> (preprint here: <https://arxiv.org/abs/quant-ph/0308039>) .

So, if Duerr Goldstein and Zanghi are correct, premise 2 is false, and the authors’ argument is unsound. I am not saying that the authors are necessarily wrong in thinking that this creates a puzzle for the Bohmians; but they need at least to recognize that their argument relies on a premise P2 which the Bohmians reject. This is a big deal and it’s an ongoing debate (see, for related debates, e.g.

<https://www.nature.com/articles/s41598-018-38261-4>

<https://www.mdpi.com/2073-8994/16/10/1325>

<https://www.nature.com/articles/s41598-024-53777-8>

<https://link.springer.com/article/10.1007/s10701-024-00798-y>)

Without this discussion, I cannot recommend this paper for publication.

This is the referee’s main point of criticism, and we agree that the presentation in the previous version of the manuscript was rather brief, which may have affected its clarity. As the referee will see in the revised manuscript, we have rewritten the text passages

addressing the Bohmian treatment of our experiment. We hope this has improved the clarity of our argument. In response to the points raised by the referee, we would like to provide the full argument for why we consider the Bohmian treatment of our experiment and its results as incorrect.

First, we would like to point out that Bohmian mechanics offers a clear description of the scenario investigated in our work. When a stream of particles encounters a semi-infinately extended potential step and the energy of the particles is insufficient to overcome the potential barrier, they are fully reflected by the step. In the steady state, this results in a standing-wave pattern with full contrast in front of the step and a non-oscillatory, exponentially decaying wavefunction within the step potential. The eigenfunction describing this scenario can be chosen to be purely real-valued; for example, a solution proportional to $\exp(-x/\lambda)$. The absence of phase gradients indicates the absence of net particle currents in the system. According to the guiding equation of Bohmian mechanics, the velocity of the particles is zero both in front of and within the step potential. In other words, the particles remain at rest throughout the system. Note that this behaviour is different from that in a potential barrier of finite width, where Bohmian mechanics predicts that the particles within the barrier have a finite velocity due to the phase of the wave function not being constant. The canonical interpretation of quantum mechanics, on the other hand, does not make any direct statements about how fast particles move within the step potential. Moreover, the usual velocity definitions in quantum mechanics, such as group or phase velocity, do not give physically meaningful results in this regime. What is clear, however, is that the motion of the particles is non-directional.

In summary, evanescent wavefunctions of the form $\exp(-x/\lambda)$, as they can occur at a step potential, imply the absence of motion in Bohmian mechanics, while in the canonical interpretation of quantum mechanics, such states only imply non-directionality or the absence of a net particle flow. Any statement beyond this requires additional physical reasoning, which we aim to provide through the proposed experimental scheme based on the population dynamics in a coupled waveguide system.

The general idea is that we compare one motion to another one that takes place simultaneously and that we understand well: the translational motion in the waveguide is compared to the hopping between waveguides. Through this comparison we conclude that the particles inside the step potential do not sit still but rather follow a speed relation $v = \sqrt{2|\Delta|/m}$ where Δ is the difference of the energy of the particle in front of the step and the potential energy inside the step. As independent evidence for this energy-speed relation, we investigate the dwell time in the system, which is shown to be well approximated by $\tau_d = \lambda/v$ where λ is the decay length of the wavefunction (corresponding to a typical travel distance into the step potential) and v is the found energy-speed relation.

In the Bohmian picture, the situation described above is plainly contradictory. On the one hand, the guiding equation postulates that the particles are at rest. On the other hand, the population dynamics in the coupled waveguides suggest that the particles are moving and that their motion is dependent on their energy. Furthermore, even if these

two statements could somehow be reconciled, there remains a second issue in the Bohmian description of the experiment related to the dwell time.

The dwell time is defined as the ratio of the particle number in the step potential to the incident particle flux $j = \hbar k_0/m$:

$$\tau_d = \left(\frac{\hbar k_0}{m}\right)^{-1} \int_0^\infty dx |\psi(x)|^2 .$$

The definition of the dwell time is based on the idea that, in a steady state, the storage time of particles in a reservoir should be given by $\tau=N/j$, where N is the number of stored particles and j is the input (or output) particle flux. In the Bohmian picture, the particles inside the step potential are not moving. In other words, they remain stored in the step potential for indefinite periods. This is consistent with the above definition of the dwell time, as, in the Bohmian picture, the particles are at rest not only inside but also in front of the potential step, causing the incoming particle flux to vanish and the dwell time to diverge. Thus, according to our understanding, Bohmian mechanics predicts a different dwell time at a semi-infinitely extended step potential compared to standard quantum mechanics (i.e., quantum mechanics without the guiding equation).

Experimentally, we can rule out dwell times that are on the same order as, or even much larger than, the lifetime of the particles in our system (260 ps). Such long dwell times would result in significant particle loss (because of mirror transmission) during the scattering process at the potential step, which is not observed experimentally. For this reason, we conclude that the guiding equation does not correctly predict the temporal characteristics of scattering at a semi-infinitely extended step potential.

Having outlined our main lines of reasoning, we would now like to comment on a point raised by the Referee: many proponents of Bohmian mechanics claim that Bohmian mechanics makes the same predictions as standard quantum mechanics (even in principle). In our understanding, such claims are based on the fact that the predicted probability densities are identical in both theories, provided the initial conditions of the Bohmian trajectories are chosen according to the Born rule (quantum equilibrium hypothesis). The latter is primarily a mathematical fact. However, we do not believe that agreement in the density is sufficient to justify the broader claim that the two theories make identical predictions.

The Referee mentions one class of examples where this issue is debated, namely arrival times. In our work, we provide another example: dwell times in scattering processes. As discussed above, the guiding equation of Bohmian mechanics predicts dwell times that are incompatible with both standard quantum mechanics and experimental results.

At a more fundamental level, the reason Bohmian mechanics deviates from the predictions of standard quantum mechanics in the described situation is that the Bohmian guiding equation does not properly account for states of non-directional motion other than the state of rest. Non-directional motion is generally represented by $v=0$ in Bohmian mechanics. This is sufficient to capture the associated net particle flux and ensures the correct probability density distribution under the action of the

guiding equation. However, it does not necessarily represent the actual temporal characteristics of a process, as was discussed above.

R1.5 This is a big deal and it's an ongoing debate [...]

We agree with the Referee that determining whether the known interpretations of quantum mechanics are equivalent is a major task in the foundations of physics. This was a key motivation for our work. We hope that the changes made to the manuscript, based on the exchange with the referees, make our line of reasoning clearer.

Referee #2

R2.1 This is a beautiful experiment with very compelling data. There is a strong case for the system being a valuable contribution to the ongoing discussion of tunnelling dynamics.

We are very pleased with this assessment by the Referee and are delighted that our work is being appreciated in this way.

R2.2 The link to Bohmian mechanics is, in my view, a distraction. Bohmian mechanics is notoriously adaptable to whatever is thrown at it, and this particular aspect of Bohmian mechanics is not particularly well developed.

The authors acknowledge that the referees (and, more broadly, the readers) may prioritize different aspects of our work. In writing this manuscript, we aimed to address both the implications of our findings for the tunnelling time debate and their relevance to deeper foundational aspects of quantum mechanics. We believe it would be a loss to remove any aspect from the paper. As the Referee will see in the resubmitted manuscript, we have expanded the discussion on Bohmian mechanics rather than omitting it, in accordance with the advice of Referee 1. We hope that the discussion now provides greater insight and is more rewarding than in the previous version, making its value easier to recognize.

R2.3 The main discussion that I was able to dig up on the hydrodynamics of Bohmian trajectories [Bittner et al., *The Journal of Chemical Physics* 112 (2000)] in tunnelling scenarios is not referenced here, although clearly that would have been very relevant.

While the reference provided by the Referee examines tunnelling in a double-well potential as an example, its primary focus is on a computational method inspired by the concept of particle trajectories in Bohmian mechanics. Nevertheless, we appreciate the Referee's point and have added additional references to works that specifically address tunnelling times from the Bohmian perspective [Leavens (1990), Spiller (1990)]. Additionally, we would like to highlight that the reviews by Landauer & Martin(1994) and Field (2022) offer further references relevant to the topic, covering both conventional quantum mechanics and Bohmian mechanics.

Referee #3

R3.1 This is a highly original and impressive experiment, and I believe the observations can contribute to the ongoing elucidation of the tunneling-time problem. The methodology appears sound and the paper places itself in the appropriate context, citing the essential prior work so far as I can tell.

We are very pleased with this general assessment of our work and thank the Referee for these kind words.

R3.2 There is one central point which gives me pause, and although I'm prepared to give the authors the benefit of the doubt that the current treatment is correct, I believe that before publication this point needs to be more rigorously supported in the text so that readers like myself feel confident in the reasoning rather than merely trusting the authors. This is the discussion around lines 69-73 about the dependence on the build-up of population in a second waveguide, and its dependence on the speed. I would expect that for two wells or waveguides coupled with a coupling constant J_0 , and an initial amplitude of A_1 in well 1 and 0 in well 2, the amplitude to be in well 2 would initially grow as a function of time as $J_0 \cdot A_1 \cdot t$, independent of the velocity.

For the case of two coupled wells, hosting localized quantum states, we agree. The time evolution for the initially unpopulated well is given by $\psi_2(t) = A_1 \sin(J_0 t)$, which is approximated by the expression provided by the Referee for small times. The Referee's comment made us realize that we did not specify in our discussion that the initial amplitude was set to 1 (or, that we are considering relative populations). This omission has been corrected in the resubmitted manuscript. The case of two coupled waveguides is addressed in the reply to the next point.

R3.3 I would ask the authors to explain in much more detail the origin of their $J_0 \cdot x/v$ prediction, and in particular whether it is valid both in cw and pulsed configurations, or only in one.

During the planning of this work, we conducted theoretical modelling to better understand how to perform the experiments and interpret the results [Klaers et al., *Phys. Rev. A* **107**, 052201 (2023)]. To address the Referee's request, we will highlight some key points from this analysis. Furthermore, we have included a more formal discussion of our experimental scheme in the Methods section of the article. This makes the manuscript fully self-contained.

A more formal justification of the experimental method can be based on modelling the coupled waveguide system using the following set of steady-state Schrödinger equations (only valid in quasi-cw configuration, not pulsed):

$$E\psi_m = -\frac{\hbar}{2m} \frac{d^2\psi_m}{dx^2} + V_0\psi_m + \hbar J_0(\psi_a - \psi_m), \quad (\text{M1})$$

$$E\psi_a = -\frac{\hbar}{2m} \frac{d^2\psi_a}{dx^2} + V_0\psi_a + \hbar J_0(\psi_m - \psi_a), \quad (\text{M2})$$

where $\psi_{m,a}$ represent the wave functions in the main and auxiliary waveguides, E is the total energy, V_0 is the potential energy, and J_0 is the coupling constant between the waveguides.

An ansatz for the solution of these equations can be written as:

$$\psi_m \propto \cos(k_1 x) e^{ik_2 x}, \quad (\text{M3})$$

$$\psi_a \propto \sin(k_1 x) e^{ik_2 x}. \quad (\text{M4})$$

This ansatz describes the motion of particles as plane wave, $\exp(ik_2 x)$, which, in case where $E < V_0$, transitions into an evanescent solution.

The relative population of the waveguides is determined by the population transfer factors $\cos(k_1 x)$ and $\sin(k_1 x)$. Using the ansatz provided above, the increase in population in the auxiliary waveguide can be shown to follow:

$$p_a(x) = \frac{|\psi_a|^2}{|\psi_m|^2 + |\psi_a|^2} \approx |k_1|^2 x^2, \quad (\text{M5})$$

for small x . Thus, a parabolic increase in the relative population in the auxiliary waveguide is expected, regardless of the type of motion. The coefficient $|k_1|^2$ is experimentally accessible.

By applying the ansatz, eqs. (M3-M4), to the coupled Schrödinger equations, one finds that k_1 and k_2 satisfy the relation:

$$k_1 = \frac{mJ_0}{\hbar k_2}. \quad (\text{M6})$$

This relation holds irrespective of whether the solution describes a propagating or evanescent state. It forms the basis of our measurement scheme, as it connects the motional state (k_2) with the relative population in the waveguides (k_1), our measurement quantity.

To justify the method, it is not strictly necessary to utilize the full information provided by eq. (M6). The key point of interest is that k_1 is proportional to J_0 (given that k_2 is independent of J_0 , which holds true for $|E - V_0| \gg \hbar J_0$, see Klaers et al., *Phys. Rev. A* **107**, 052201 (2023)). This relationship is, in principle, already evident from the coupling between two localized states, as discussed in the article.

As J_0 is a property of our measurement device and not of the state of motion, we factor it out as a pre-factor, $|k_1|^2 = (J_0/v)^2$, resulting in:

$$p_a(x) \approx |k_1|^2 x^2 = \left(\frac{J_0}{v}\right)^2 x^2. \quad (\text{M7})$$

The remaining factor v has the dimensions of a velocity and an undetermined sign. Most importantly, this quantity determines the spatial length scale of the population build-up in the coupled waveguide system—an effect that genuinely represents a spatio-temporal phenomenon, distributing particles in space (x) relative to a temporal reference (J_0). We regard this as a property that can only be attributed to a form of motion.

R3.4 My impression is that they assume the coupling begins abruptly at $x=0$, <...>

Yes, this is an assumption we make in our analysis. In the experiment, the coupling is introduced more gradually, as the structures confining the light have a finite lateral resolution. We estimate the uncertainty in the starting point—where the coupling 'begins'—to be on the order of $2 \mu\text{m}$. In the Methods section, we investigate the systematic errors in the experiment, including the one raised by the Referee. A $2 \mu\text{m}$ shift in this quantity results in a systematic change of approximately 8.5% in the extracted speeds.

R3.5 <...> and that (at all instants t), the population in waveguide 2 at a given position x involves the coherent sum of population already present in waveguide 2 at $x-\Delta x$ and the population present in waveguide 1 at x , such that the effective coupling time is x/v . On the one hand, this seems plausible, <...>

We consider this to be essentially correct. In our presentation of the method in the main paper, after discussing two coupled (localized) quantum states, we introduce the coupled waveguide system, where we essentially replace the time t with x/v . We believe this is physically plausible. However, this argument does require additional justification, which we have provided above.

R3.6 <...> but especially given their desire to draw conclusions about the average v^2 in cases where the phase is constant and the Bohm description involves no motion at all, I would have liked to hear them comment on the dependence on the sign of v . They give a formula for the probability, not the amplitude, which depends only on v^2 ; <...>

We believe it is no coincidence that our measurement scheme does not yield a sign. A sign implies directionality, indicating net particle transport. Any measurement scheme sensitive to direction must necessarily yield a velocity of zero for an evanescent state of the form $\exp(-x/\lambda)$, since such states do not imply net particle transport. Conversely, any scheme that assigns a non-zero speed value to a state of this form, which is true for the scheme presented in this work, must be insensitive to direction.

R3.7 <...> but if there is a superposition of backward- and forward-propagating waves, does the intuitive argument about interference in the forward direction actually go through? Or have I misunderstood their argument from the start?

The question of whether an evanescent wave function of the form $\exp(-x/\lambda)$ can be thought of as superposition of forward- and backward-propagating waves is both interesting and likely a physically deep one. A similar question arises in the tunnelling time debate, where it is asked whether the wavefunction inside a barrier can be interpreted as a superposition of a wavefunction corresponding to reflected particles and one corresponding to transmitted particles. We do not believe this is possible in either scenario, but we must acknowledge that this statement requires a solid argumentative foundation, which we are unable to provide at this time.

R3.8 Is it even clear that this is the relevant timescale, and that the buildup is not determined by the duration of the entire pulse, for instance?

The cavity mirrors used in this experiment provide extremely high reflectivity. Even with the relatively small mirror separations we employ (of order 10 μm), we achieve high photon lifetimes of 260 ps. This leads to a clear separation of timescales: the duration of the scattering process (dwell time) is 1–2 ps, the photon lifetime is 260 ps, and the optical pumping lasts 26 ns. Since the photon lifetime is much longer than the dwell time, the finite lifetime of the particles has no measurable influence on the scattering. Conversely, because the photon lifetime is much shorter than the pulse duration, it can be ruled out that the buildup of population in the cavity occurs over, or is determined by the duration of, the entire pulse.

R3.9 If the authors can provide a satisfactorily clear and (semi-)rigorous justification for the interpretation of their coupling data in terms of a velocity, then I would recommend this paper for publication.

We thank the Referee for their valuable comments, observations, and insights on our experiment and manuscript. We hope that our revisions—particularly the more formal justification of our measurement scheme provided in the Methods—have adequately addressed the Referee’s concerns.

Response to Referees – April 28, 2025

Manuscript: Sharoglazova et al., Energy-speed relationship of quantum particles challenges Bohmian mechanics, 2024-05-10504C

We thank the Referees once again for their reports and are grateful that both acknowledged a significant improvement in the presentation of our work. While both Referees noted that the manuscript could still benefit from further clarification, we found it challenging to present our results more clearly and in greater detail without exceeding the word limit for a *Nature* article and without sacrificing other crucial content, such as the more technical aspects of our experiment. In fact, to the best of our understanding, the previous version of the manuscript already exceeded the word limit. To address this, the Editor kindly granted us an additional 400 words, allowing us to present our results in a less condensed format while preserving the overall structure of the manuscript. We believe this revision adequately addresses the concerns regarding presentation raised by the Referees.

Furthermore, following a suggestion from the Editor, we have revised the title of our manuscript. The new title is: “Energy-speed relationship of quantum particles challenges Bohmian mechanics.” The Editor suggested that the title should ideally not only set the stage but also indicate the main result of the work. We agree that the new title does this more effectively than the previous one.

The main point addressed by Referee 1 is that the connection of the paper to Bohmian mechanics should still be improved. The Referee mentions a specific example:

(Even if they could have explained their position better. In fact, in the response letter they write: “we do not believe that agreement in the density is sufficient to justify the broader claim that the two theories make identical predictions.” But why not? What would they need more than that? Isn't the only thing (probability density) that quantum mechanics provide to connect the formalism with the experiments? They would need to explain better to convince anyone of this).

In the revised manuscript, we have expanded the discussion of this issue using the example of the dwell time. Although the dwell time can be defined identically in both theories, we believe it differs between them. Most of the additional text granted by the Editor is indeed dedicated to addressing this point.

Referee 2 suggests that we embed our work more strongly within the existing debate in the literature, citing a specific example from the review paper by Fields. The passages referred to specifically concern tunnelling times, and therefore, not all aspects of that debate are directly relevant to our work. For instance, there is no consensus within standard quantum mechanics on how to calculate the tunnelling time (as opposed to the dwell time). This makes discussions about using tunnelling times to distinguish between different theories somewhat speculative. We believe that this speculative nature is, to some extent, reflected in the cited passages.

At the same time, these passages also reflect the widely held conviction among many authors that Bohmian mechanics should be equivalent to standard quantum mechanics in all measurable quantities. We therefore interpret the Referee's request for a broader embedding in the literature debate as essentially a call for a more detailed explanation of the claimed non-equivalence. To a large extent, this aligns with the request made by Referee 1 in their report, and we believe that the changes made to the manuscript effectively address the concerns raised by both Referees.

In addition to addressing the points raised by the Referees, we have corrected an inaccuracy in the data analysis related to the dwell time. Our determination of the dwell time is based on measuring both the number of stored particles and the ingoing particle current. The ingoing current is derived from the analysis of standing wave patterns in front of the potential step, which we analyse by identifying the maximum density within that pattern. However, the determination of this maximum was not entirely accurate, as the domain in which our data analysis script searched was slightly too limited. This issue has now been corrected giving rise to minor corrections in the reported dwell time. The interpretation of our results, however, remains unchanged. Furthermore, we have added more detailed information in the manuscript regarding how the dwell time is calculated from our experimental data.

To facilitate the review process, we have added a version of our manuscript in which all changes are marked in red.

We would once again like to thank the Referees and the Editor for their valuable input, which has helped us improve the clarity of our manuscript and, we hope, enhance its impact on the ongoing scientific debate. While the discussion of tunnelling times and Bohmian mechanics can at times be controversial, we consistently felt that the Referees' feedback constructively challenged us to present the reasoning behind our experiment as clearly and transparently as possible.